# Hydrodynamic Behavior of a Pump as Turbine under Transient Flow Conditions

**Jianxin Hu** [1]**, Xianghui Su** [1,*]**, Xin Huang** [1]**, Kexin Wu** [1]**, Yuzhen Jin** [1]**, Chunguang Chen** [2] **and Xulai Chen** [2]

[1]  Faculty of Mechanical Engineering and Automation, Zhejiang Sci-Tech University, Hangzhou 310018, China; jhu@zstu.edu.cn (J.H.); dx.huang@outlook.com (X.H.); kexin.wu@zstu.edu.cn (K.W.); gracia1101@foxmail.com (Y.J.)
[2]  Zhejiang Testing&Inspection Institute for Mechanical and Electrical Products Quality Co., Ltd., Hangzhou 311300, China; c330300206@163.com (C.C.); chenxulai945@163.com (X.C.)
[*]  Correspondence: suxianghui@zstu.edu.cn

**Abstract:** Centrifugal pumps as turbines (PATs) are widely used in chemical engineering for recycling the abundant energy from high-pressure fluid. The operation of PATs is significantly affected by their upstream conditions, which are not steady (i.e., with a constant flow rate) in reality, thus, research on the flow mechanism of PATs under transient conditions should be considered of higher importance. In this study, a numerical model of a PAT was developed by employing the sliding mesh method to describe turbine rotation, and a user-defined function (UDF) for characterizing transient flow conditions. Corresponding experiments were also conducted to provide validation results for the simulation, and the simulation results agreed well with the experimental outcomes. The instantaneous characteristic curves under the current working conditions were obtained for different transient flow rates. The results show that the turbine's efficiency is significantly affected by transient flow conditions. In particular, a rapid increase (large time derivative) of flow rate results in a large energy dissipation at the turbine outlet, and therefore, the turbine efficiency decreases. In addition, as the flow rate increases, the hydrodynamic force on the impeller, and the pressure fluctuation amplitude in the volute first decrease and then increase, reaching the minimum near the design flow rate. The current study provides a reliable and precise approach for the estimation of hydrodynamic performance of fluid machinery under transient flow conditions.

**Keywords:** centrifugal pumps as turbines; transient flow conditions; efficiency; stability characteristics



## 1. Introduction

A pump is a reversible rotating machine that can be turned into a turbine, i.e., a pump as turbine (PAT). It can utilize the energy from a high-pressure liquid by converting the high-pressure liquid's energy into a rotating pump shaft's mechanical energy. Williams, Chapallaz et al., and Buono et al. found that PATs can be widely used in the chemical engineering industry, for small water conservancy, and in hydropower resources owing to their simple structure, low cost, stable operation, and convenient maintenance [1–3].

Because of PATs' advantages, numerous in-depth studies concerning PATs have been performed in the last few years. However, the connection between a pump's positive and negative working conditions is still a problem that must be solved. Several experimental investigations were conducted by Stefanizzi et al. [4]; then, scholars derived the best efficiency relationships to use during predictions for PATs [5–7]. However, these relationships are only valid within a specific speed range. Yang et al. applied precise numerical simulations to predict the best efficiency point (BEP) of a turbine for a centrifugal pump's positive and negative working conditions [8,9]. Bozorgasareh et al. then explored pump selection through numerical calculations, designed parameters for components such as impellers and wheel covers, and optimized the impeller model so it produced a higher efficiency and more stable working conditions [10,11]. In addition to the geometric parameters that

affect a PAT's operation, external working conditions also significantly impact its efficiency. Delgado et al. found through experimentation that different speeds often had different optimal efficiency points, and that the maximum energy could be recovered by controlling the flow or the pressure downstream of the PAT [12]. By analyzing the flow field inside the impeller, Su et al. revealed that there were also non-negligible hydraulic losses in the areas outside the rotor, such as in the volute and the cavity [13].

The various operation modes of a pump are shown in Figure 1. Mode A involves normal operation as a pump between maximum flow rate $Q_{max}$ at head H = 0; Mode B is the "brake mode" which results in fluid flowing back through the pump; Mode C involves normal operation as a turbine; in Mode D, the turbine works as a brake when the flow rate is too low; in Mode E, the pump works with an abnormal sense of rotation due to the pressure in the discharge nozzle dropping further; in Mode F, the pump works as a brake when the pressure in the discharge nozzle drops below the pressure prevalent in the suction nozzle; Mode G involves the reverse flow turbine, because the pressure in the suction nozzle is sufficiently greater than the pressure in the discharge nozzle; Mode H is operation between the range where the rotor can deliver power as a turbine, and H = 0 [14].

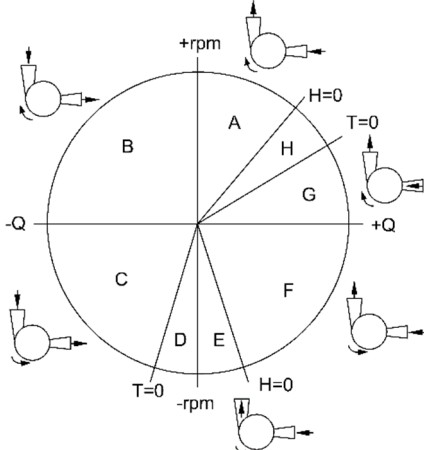

**Figure 1.** Operation modes.

To investigate the starting and stopping conditions for a centrifugal pump's forward rotation, Wu et al. used the sliding mesh method to study a pump's performance by developing a method to simulate the flow field during the start-up process, using an interface to connect the rotating impeller and the stationary parts [15]. Farhadi et al. analyzed the water hammer phenomenon that occurred during the start-up process [16,17]. They specifically investigated the acceleration effect when the flow increased, as well as the vortex rotation's transient effect. Chalghoum et al. studied the transient characteristics of a centrifugal pump during the starting period, and found that the pressure evolution is influenced by the valve opening percentage and starting time [18]. To establish stopping conditions, Feng et al. studied the transient process of centrifugal pumping during stopping experiences, mainly in four modes, namely, pump mode, breaking mode, turbine mode, and runaway mode [19].

Scholars have conducted many studies regarding PATs, but much of this work has been based on research and analysis under stable conditions. In reality, PATs are often subject to upstream traffic; in this situation, their flow rates increase or decrease over time. Therefore, the existing research concerning PATs should be supplemented with an analysis of the impact of variable working conditions on PATs (in terms of turbine operation). To address the shortcomings of the existing research, this study used the commercial software FLUENT to perform numerical simulations to obtain performance characteristic curves that match the experimental data [20]. After verifying the numerical method's feasibility, the transient flow case of the PATs corresponding to Mode C in Figure 1, was simulated using a user-defined function (UDF) and the sliding mesh method [21]. The UDF was used

to simulate the inlet conditions with different increasing rates of flow, and the sliding mesh method was used to simulate the rotational motion of the rotor. The difference between the performance characteristic curves of transient and steady-state operating conditions, and the change in the PAT's working state due to increases in flow rate were studied by analyzing internal flow fields and stability characteristics.

## 2. Numerical Methods, Models, and Validation

### 2.1. Governing Equations

The fluid conformed to the mass conservation, momentum conservation, and energy conservation laws via the corresponding equations. The PAT's interior fluid flow was regarded as an incompressible, three-dimensional, unsteady turbulent flow. Heat transfer was neglected during the current study, and the energy conservation equations were not considered. The mass and momentum conservation equations are presented in Equations (1) and (2), respectively:

$$\frac{\partial \rho}{\partial t} + \frac{\partial}{\partial x_i}(\rho u_i) = 0, \tag{1}$$

$$\frac{\partial}{\partial t}(\rho u_i) + \frac{\partial}{\partial x_j}(\rho u_i u_j) = \frac{\partial}{\partial x_j}\left(\mu \frac{\partial u_i}{\partial x_j}\right) - \frac{\partial p}{\partial x_i} + S_i. \tag{2}$$

In Equations (1) and (2), $\rho$ and $\mu$ are the fluid's density and dynamic viscosity, respectively, $u_i$ is the velocity in the $i$ direction, $p$ is the hydrostatic pressure, and $S_i$ is the generalized source term.

Regarding the choice of turbulence model, previous studies have shown that the $k - \varepsilon$ model was suitable for investigations that focused on hydrodynamics, whereas the $k - \omega$ model was more appropriate when flow-field characteristics were emphasized. When both hydrodynamic and flow-field characteristics must be considered, adopting the $k - \omega\ SST$ model and a small value of $y^+$ for the first boundary layer should be suitable and have sufficient accuracy [22,23]. Although a large eddy simulation (LES) would have a higher calculation accuracy and could accurately reflect the characteristics of the flow field, it would require a high grid quality and a significant amount of computing resources. To conserve computing resources as much as possible while obtaining an acceptable calculation accuracy, the $k - \omega\ SST$ turbulence model was chosen for this study. The $k - \omega\ SST$ turbulence model has an accuracy associated with near-wall viscous flow, as well as a reliability related to calculations for far-field free flow. This model considers the turbulent shear stress, and can obtain accurate results even when calculating flow separation. $k$ and $\omega$ could be determined by solving Equations (3) and (4):

$$\frac{\partial(\rho k)}{\partial t} + \frac{\partial(\rho k u_i)}{\partial x_i} = \frac{\partial}{\partial x_j}\left[(\mu + \frac{\mu_t}{\sigma_k})\frac{\partial k}{\partial x_j}\right] + G_k - \rho k \omega \beta^*, \tag{3}$$

$$\frac{\partial(\rho\omega)}{\partial t} + \frac{\partial(\rho\omega\overline{u_i})}{\partial t} = \frac{\partial}{\partial x_j}\left[(\mu + \frac{\mu_t}{\sigma_\omega})\frac{\partial\omega}{\partial x_j}\right] + \frac{\partial\omega}{\partial k}G_k - \rho\omega^2\beta + 2(1 - F_1)\rho\frac{1}{\omega\sigma_\omega}\frac{\partial k}{\partial x_j}\frac{\partial\omega}{\partial x_j}. \tag{4}$$

The turbulence viscosity, which appears in Equation (2), could then be calculated using Equation (5):

$$\mu_t = \frac{\rho k}{\omega}. \tag{5}$$

Among them, $G_k$ is the turbulent kinetic energy due to the average velocity gradient, $k$ is the turbulent kinetic energy, $\omega$ is the specific dissipation rate.

### 2.2. Numerical Models

The PAT's numerical model, presented in Figure 2, consisted of six parts: inlet and outlet extensions, front and back chambers, a volute, and an impeller (which included

six balance holes). The length of inlet extension and outlet extension was determined by the monitoring position in the experiment. The meshes of the PAT's fluid domain were generated using the meshing software ICEM. As shown in Figure 3, the computational domain was constructed using hexahedral grids. Grids close to the impeller's near-wall surface were refined to ensure that the first boundary layer's $y^+$ value was less than 50 [20].

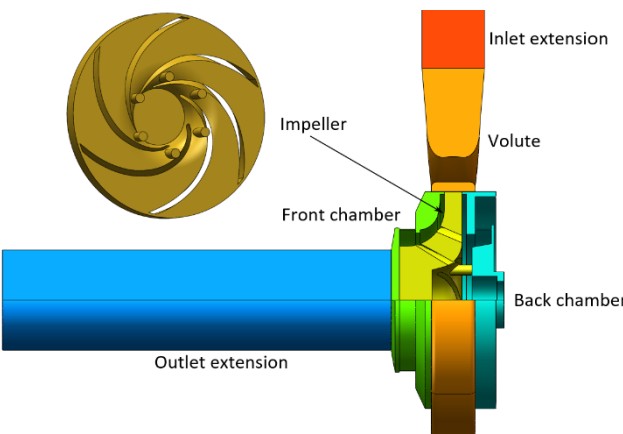

**Figure 2.** Numerical model of a pump as turbine (PAT).

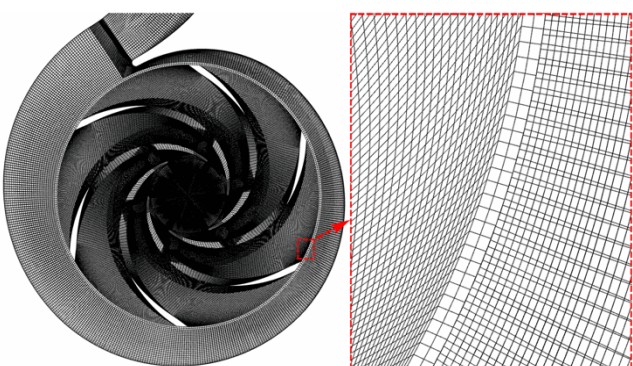

**Figure 3.** Hexahedral grids in the impeller's computational domain (**left**), and a zoomed-in view of the area around the volute (**right**).

The independence of the grids with different grid numbers was verified first. The simulation results which were built on the same test case in steady operation, when the PAT operated under the design flow rate (80 m$^3$/h) and the design speed (2900 rpm), are shown in Figure 4. These results show that when there were 6.12 million grids, the efficiency gap was less than 0.5%; this indicates a high calculation accuracy and the conservation of calculation resources. In addition, the conclusion of the grids' independence is also suitable for other working conditions and transient conditions.

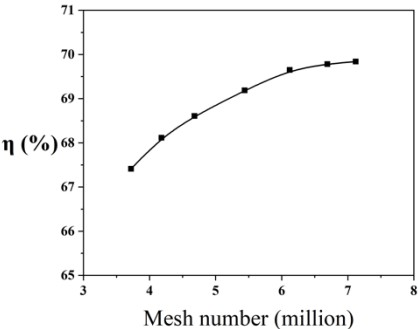

**Figure 4.** Effect of mesh number on efficiency.

### 2.3. Setup of The Numerical Simulation

When setting the boundary conditions, the pressure inlet was selected as the inlet boundary, and the mass flow outlet was chosen as the outlet boundary. The wall's surface roughness was set to 50 μm, which is consistent with the roughness of the test machine.

When setting the steady working conditions, the use of the motion reference frame (MRF) method to simulate the relative rotation between the runner and the volute enabled data exchange between the interfaces. Performing transient calculations based on the results of steady calculations can stabilize the calculation results more quickly. When setting the conditions for the transient calculation examples, the MRF method could not be applied, so the rotor rotation was simulated using the sliding mesh method. If the impeller's rotation angle corresponding to each time step was 1–2°, the calculation results would have a high accuracy [24]. Therefore, for a rotational speed of 2900 rpm, the time step was set to $5.8 \times 10^{-5}$ s. The number of iterations for each step is 20, and the convergence criterion of the residual was less than $10^{-5}$ s. For transient flow conditions, keeping the rest of conditions constant, a UDF was used to control the flow rate at the inlet.

### 2.4. Validation of The Numerical Simulation

An experimental test was conducted to validate the numerical simulations. The layout of the test system, shown in Figure 5, included a light vertical multistage centrifugal pump placed upstream for pressurization. Its model number was CDL150-40-2FSWPC, and it was manufactured by Nanfang Pump Industry Co., Ltd. This pump's power can reach 45 kW, and its working head and flow rate were 70.5 m and 150 m³/h, respectively. The booster pump provided high-pressure fluid for the turbine. The central console could be used to adjust the inlet pressure and the flow rate. The turbine was connected directly to the eddy current dynamometer by a flexible coupling. The eddy current dynamometer's load could be adjusted to ensure that the turbine maintained the same speed at different flow rates. Pressure transmitters were arranged at the inlet and outlet of the turbine, and an electromagnetic flowmeter was also installed at the inlet.

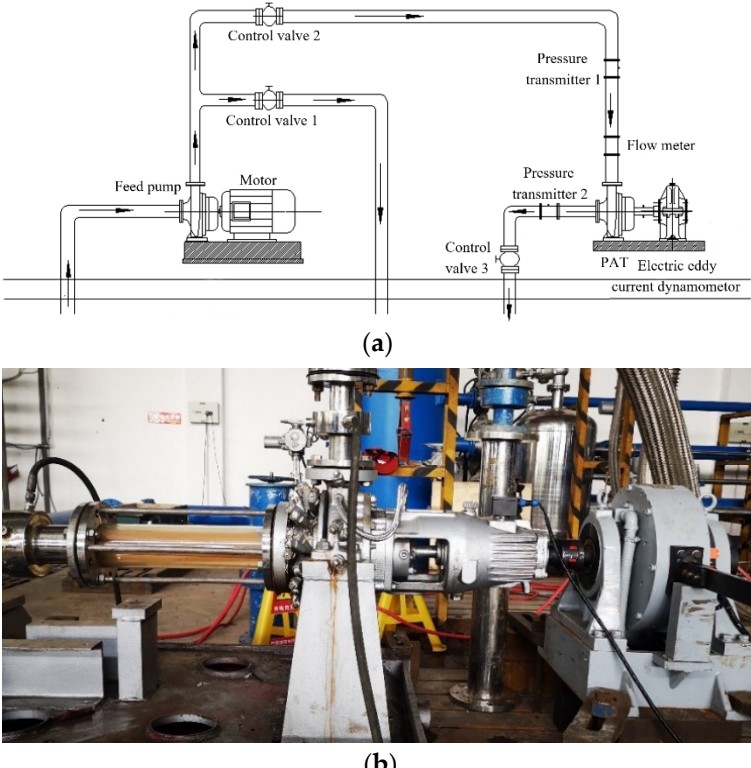

(**a**)

(**b**)

**Figure 5.** Schematic of the current experimental system (**a**), and a picture of this system in the lab (**b**).

The measuring range of the flow meter was 20–200 m³/h, while the accuracy was $\pm0.5\%$. The measuring range of the pressure transmitter was $-100$ kpa–10 Mpa, while the accuracy was $\pm0.1\%$. The measuring range of the speed sensor was 0–1300 rpm, while the accuracy was $\pm0.5\%$, and measuring range of the torque sensor was 0~100 Nm, where the accuracy was $\pm0.5\%$. The first-order uncertainly analysis adopted the constant odds combination method. The uncertainty of the flow rate, head, shaft power, and efficiency were $\pm1.44\%$, $\pm1.04\%$, $\pm2.12\%$, and $\pm2.72\%$, respectively.

The primary object investigated in the current study was a single-stage single-suction cantilever turbine, as shown in Figure 6. The classical pump parameters were adopted, and its related parameters and runner diagram are given in Table 1 and Figure 7, respectively. Table 2 displays the operating parameters at BEP.

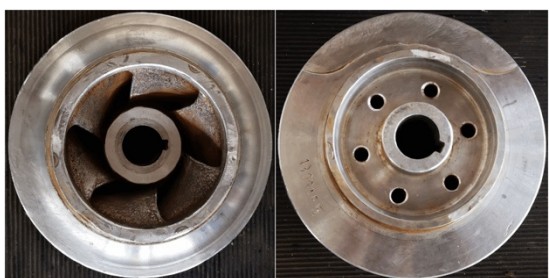

**Figure 6.** Pictures of the current study's PAT. **Left**: front chamber of the impeller; **right**: back chamber of the impeller.

**Table 1.** Geometric parameters and performance parameter of the PAT.

| Parameters | Notation | Value |
|---|---|---|
| Impeller inlet diameter (mm) | $D_1$ | 169 |
| Impeller outlet diameter (mm) | $D_2$ | 86 |
| Blade inlet width (mm) | $b_1$ | 14 |
| Blade outlet width (mm) | $b_2$ | 26 |
| Blade inlet angle (°) | $\beta_1$ | 30 |
| Blade outlet angle (°) | $\beta_2$ | 15 |
| Blade wrap angle (°) | $\phi$ | 142 |
| Number of blades | $Z$ | 6 |
| Specific speed (pump operation) | $N_{sd}$ | 90 |

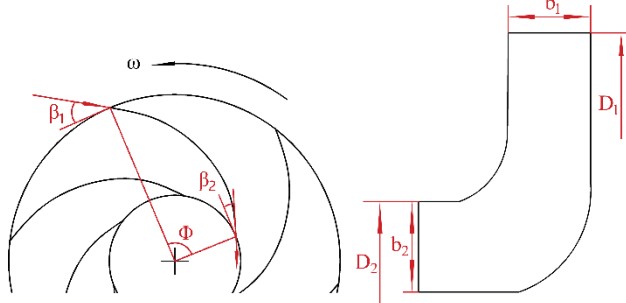

**Figure 7.** PAT runner diagram.

**Table 2.** PAT operating parameters at the best efficiency point (BEP).

| BEP Parameters | Notation | Value |
|---|---|---|
| Flow rate (m³/h) | $Q_d$ | 80 |
| Head (m) | $H_d$ | 54 |
| Speed (rpm) | $N_d$ | 2900 |

The characteristic parameters $P_{out}$, $P_{in}$, and $H$ are expressed in Equations (6)–(8), respectively:

$$P_{out} = MN, \tag{6}$$

$$P_{in} = Q(p_{in} - p_{out}), \tag{7}$$

$$H = \frac{p_{in} - p_{out}}{\rho g}. \tag{8}$$

Among them, $Q$ represents the flow rate, $p_{in}$ and $p_{out}$ are the total pressures at the PAT's inlet and outlet sections, respectively, $M$ denotes the moment on the impeller, and $N$ is the impeller's angular velocity.

The dimensionless parameters were used to investigate the PAT performance, and are expressed by Equations (9)–(12), where $\phi$ is the flow number, $\psi$ is the head number, $\pi$ is the shaft power number, and $\eta$ represents the efficiency of PAT.

$$\phi = \frac{Q}{ND_1^3}, \tag{9}$$

$$\psi = \frac{gH}{N^2 D_1^2}, \tag{10}$$

$$\pi = \frac{P_{out}}{\rho N^3 D_1^5}, \tag{11}$$

$$\eta = \frac{\pi}{\phi\psi}. \tag{12}$$

Figure 8 shows the PAT's characteristic curves generated from the experimental results and numerical simulations at the design speed (2900 rpm). According to the curves, the numerical simulation results agreed with the experimental results very well. Specifically, the differences between numerical and experimental values in the energy efficiency, head number, and output power number at the design flow rate (80 m³/h) were 2.4%, 1.5%, and 2.2%, respectively. The numerical results were slightly larger than the experimental results, mainly due to mechanical and volume losses during the experiments, which were not taken into account in the numerical simulations. Meanwile, the working state of the rotor during experimentation changes slightly at every moment, but only the data at a certain moment is recorded. In addition, the results show that the PAT's efficiency continued to increase as the flow rate increased. After the design flow rate was exceeded, the turbine's energy conversion efficiency began to decrease slowly.

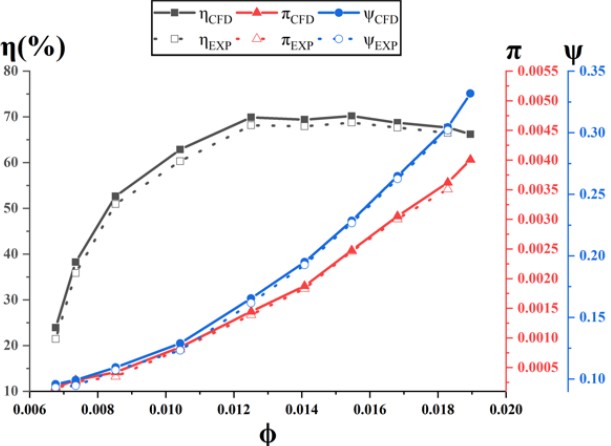

**Figure 8.** Comparison of experimental and numerical simulation results at the design speed (2900 rpm).

## 3. Results and Discussion

### 3.1. Analysis of the Characteristic Curve

The PAT's working conditions were not steady during its operation, and the flow rate and pressure conditions continued to change over time. Therefore, the PAT's energy conversion efficiency was also constantly changing under the unsteady working conditions. Chalghoum et al. concluded that the transient characteristics of a centrifugal pump during the starting period are influenced by the valve opening percentage and the starting time [18]. Wu et al. controlled the flow rate to increase linearly to simulate the rapid opening process of the pump, in which the acceleration effect of the increase in flow rate and the transient effect of the vortics revolution have significant impacst on the performance of the pump [15]. Based on the research above for the transient flow conditions, the flow was controlled to increase linearly, and the corresponding time was limited within 1 s, thereby simulating the process of rapid changes in the inlet flow. Additionally, this research compared transient flow conditions with constant flow conditions.

The characteristic curve of the linear flow valve used in the PAT system is shown in Figure 9. When the linear flow valve was adjusted, its characteristic curve and opening curve often presented a trend of upward or downward change, and its comprehensive trend was approximately linear. By controlling the valve opening to increase proportionally with time, the inlet flow rate of the PAT can be changed in an almost linear fashion. To facilitate the simulation of changes in flow, we regarded this factor as an approximately linear change in the numerical simulation.

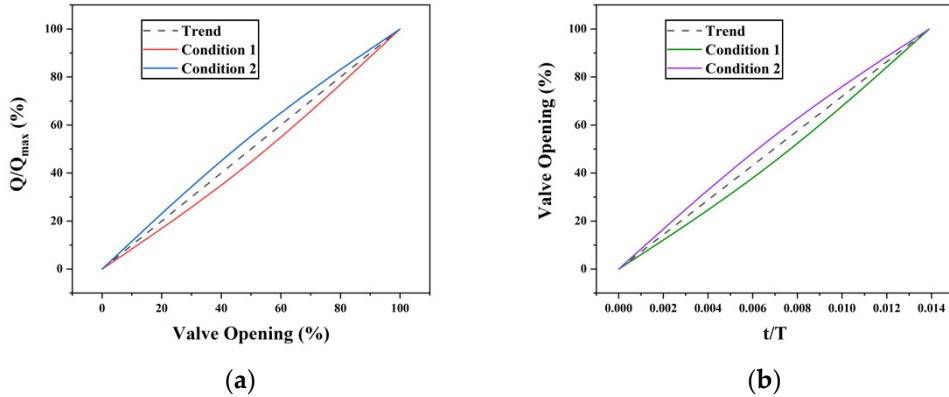

**(a)**          **(b)**

**Figure 9.** Characteristic curve of linear flow valve (**a**), and opening curve (**b**).

Table 3 shows four sets of working conditions, among which Condition 1 corresponds to a group of stable condition points. In this group, the PAT operated at a constant speed (2900 rpm), and a constant flow rate for each condition point. The average values of the PAT's pressure and torque were obtained over six cycles, and then the transient characteristic curve for the new stable working conditions group was obtained, with the flow rate as the scale. At this time, the rate of flow change was regarded as 0. Conditions 2–4 correspond to transient flow conditions under different increase rates of flow, and the flow change rule is graphically represented in Figure 10. For these three working conditions, the other set conditions were identical to those in the stable working conditions group, but the flow rate linearly increased under the operation range.

**Table 3.** Different working conditions.

| Serial Number | Time Consumed (s) | Increase Rate of Flow (kg/s$^2$) |
|---|---|---|
| Condition 1 | unlimited | 0 |
| Condition 2 | 0.696 | 35 |
| Condition 3 | 0.348 | 70 |
| Condition 4 | 0.174 | 140 |

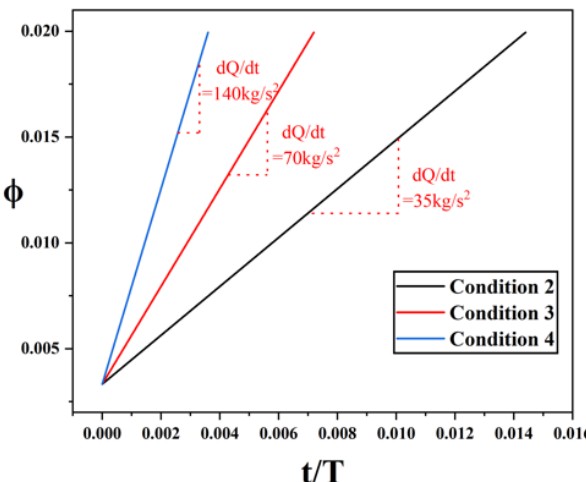

**Figure 10.** Comprehensive flow rate change law for various conditions.

The characteristic curves for the four groups of working conditions are shown in Figure 11. The characteristic curves for the three transient flow conditions were very similar, and there was a gap between them and the curve for the stable conditions. Figure 11a shows that the shaft power number for the stable working conditions was always higher than for the transient working conditions. Among them, the values of shaft power for Conditions 3 and 4 were higher than for Condition 2, when the flow rate was low. As the flow rate continued to increase, the change in torque for Condition 3 was consistent with that for Condition 2, and when the flow for Condition 4 increased to 0.875 $Q_d$, its torque became smaller than for the other working conditions. Figure 11b shows the head number change trend for each group of working conditions. Among them, the head number for Condition 2 was always lower than for the stable working conditions, and it maintained good stability, while the head number values for Conditions 3 and 4 in the low flow range were higher than for Conditions 1 and 2. As the flow increased, Conditions 3 and 4 converged with Condition 2 near the design flow and remained consistent thereafter. Figure 11c shows the efficiency curve for each set of working conditions. The error was the accumulation of the power number and the head number. As the flow rate increased, the efficiency curves for the variable working conditions gradually deviated from those of the stable working conditions. The higher efficiency curve, that is, the characteristic curve under stable working conditions, was ideal. If the flow rate changed slowly, the efficiency in the middle and high flow ranges would be higher, and would always be lower than for the stable conditions. Under transient flow conditions, the efficiency changes in the low flow range were very chaotic and had no obvious periodicity. The efficiencies for the three transient flow conditions were significantly lower than the efficiency for the stable conditions. This may have occurred because of the turbine stall phenomenon under low flow rate conditions. As the flow rate increased to 0.75 $Q_d$, the efficiency changes for the transient flow conditions began to show periodicity. Additionally, Conditions 2 to 4 began to show different efficiency changes, that is, a greater increase in rate of flow led to lower efficiencies. At $Q_d$, the efficiencies for all the conditions reached the maximum values and showed the same slow decline trends after surpassing the design flow rate. When the rate of flow increase was less than or equal to 70 kg/s$^2$, the PAT's efficiency change trend was very close to the stable conditions trend, with no large differences.

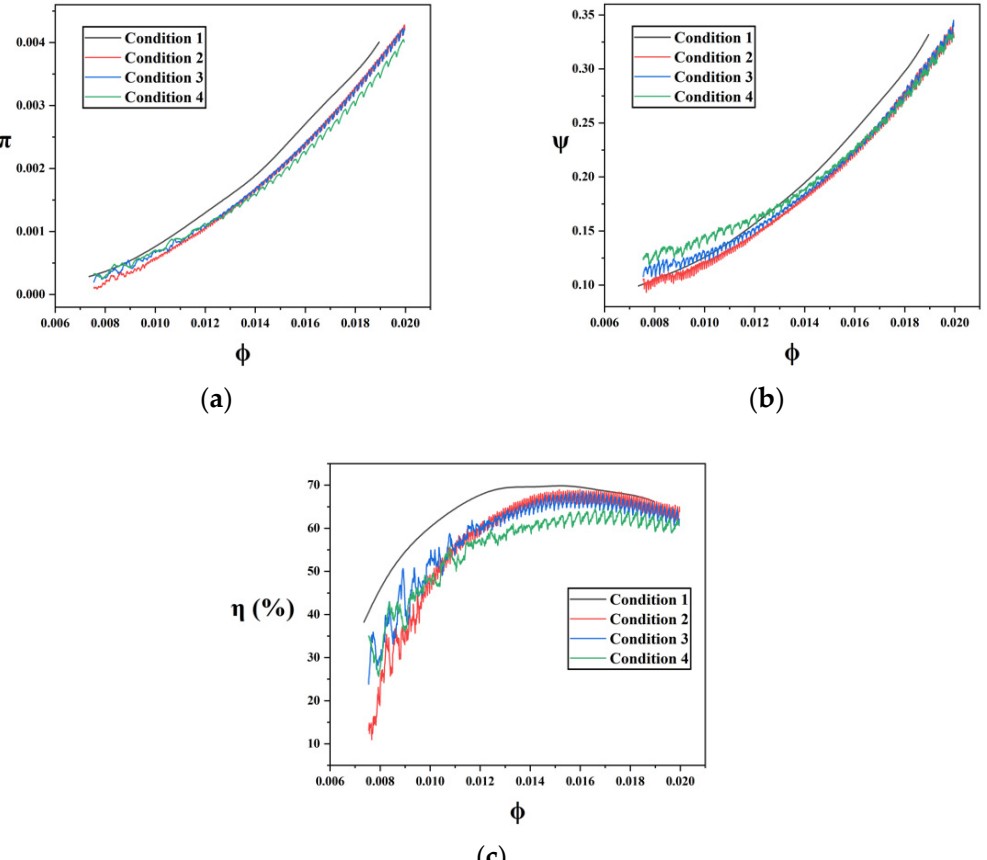

**Figure 11.** Power number curves (**a**), head number curves (**b**), and efficiency curves (**c**) for different working conditions.

### 3.2. Analysis of PAT's Internal Flow Field

The differences between the characteristic curves for the transient flow conditions and constant flow conditions were identified, which were caused by differences between the internal flow fields for each set of working conditions. An analysis and a comparison of the impeller's internal flow field for different working conditions was conducted to study the reasons for the differences between the characteristic curves.

The pressure is expressed by the pressure coefficient, which is expressed as follows:

$$C_p = \frac{p - p_{ref}}{0.5\rho u^2}. \tag{13}$$

In this equation, $p$ is the pressure at the monitoring point, $p_{ref}$ is the reference pressure, and $u$ is the peripheral speed at the impeller inlet.

A static pressure cloud diagram of the impeller channel's radial section is shown in Figure 12. As the flow rate increased, the static pressure in the impeller gradually decreased, indicating that the turbine had a relatively weak ability to transform and utilize pressure energy at low flow rates. The conversion of pressure energy in the impeller's flow channel increased as the flow rate increased, and the low-pressure area at the impeller's outlet rapidly expanded, occupying the entire flow channel after the design flow rate was exceeded.

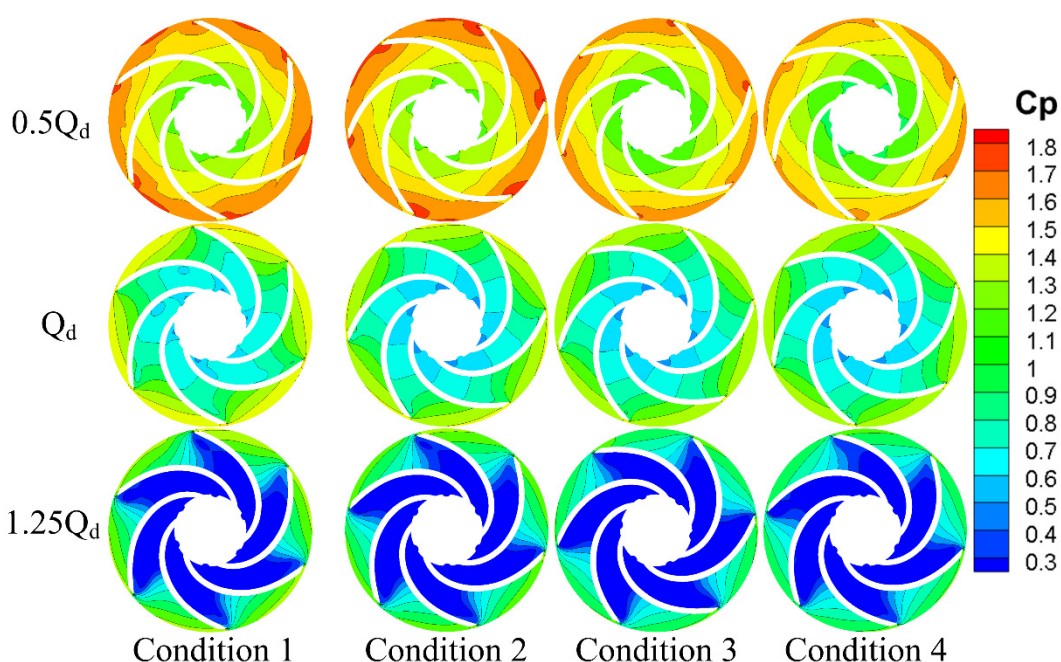

**Figure 12.** Impeller channel's radial section static pressure cloud diagram.

Next, the static pressure cloud maps for the four working conditions were compared for the same flow rate. When the flow rate was 0.5 $Q_d$, a greater increase in the rate of flow led to smaller high pressure zones at the inlet of the impeller's suction surface, as well as more densely distributed pressure contours. This indicated greater inlet and outlet pressure gradients. A larger pressure gradient indicated that there was a larger pressure difference between the inlet and outlet of the impeller. For this reason, the head numbers for Conditions 3 and 4 in Figure 11b were larger than those for Conditions 1 and 2. For the flow rates $Q_d$ and 1.25 $Q_d$, there were no obvious differences between the pressure distributions for each group of working conditions. This phenomenon was consistent with the head number change trends for the high flow rate in Figure 11b.

Figure 13 presents the turbulent kinetic energy distributions and streamline distributions for the PAT impeller's radial section. When the flow rate was 0.5 $Q_d$, the streamline distribution in the impeller was chaotic, and there were large vortices in the flow channel. There were high turbulent kinetic energy regions at the impeller's inlet. The streamlines in the impeller's flow area for $Q_d$ were uniformly distributed, without obvious high turbulent kinetic energy regions. However, there was a secondary flow near the pressure surface, which may have been caused by the fluid impacting the blades and causing the flow direction to change. A weak vortex cluster appeared at the entrance of the blade's suction surface, and there was a small area of high turbulent kinetic energy, which may have been caused by inconsistencies in the blade's placement angle and the inlet velocity. The distribution of these turbulent kinetic energy regions presented an interesting characteristic: no vortex was generated at the entrance of the blade's suction surface in the flow channel near the tongue, rather, a vortex was generated in the flow channel far from the tongue. At 1.25 $Q_d$, the streamline distribution was relatively regular. However, the weak vortex at the entrance of the suction surface expanded rapidly when the flow rate increased and occupied nearly half of the flow channel. The vortex's turbulent flow energy at 1.25 $Q_d$ was also higher than that at $Q_d$ and 0.5 $Q_d$, the transfer of fluid energy was seriously hindered, and the turbine's efficiency was affected. As shown in Figure 11c, the result of the continuous development of the vortex cluster at this location led to a gradual decrease in the turbine's energy conversion efficiency after the flow rate exceeded the design flow rate.

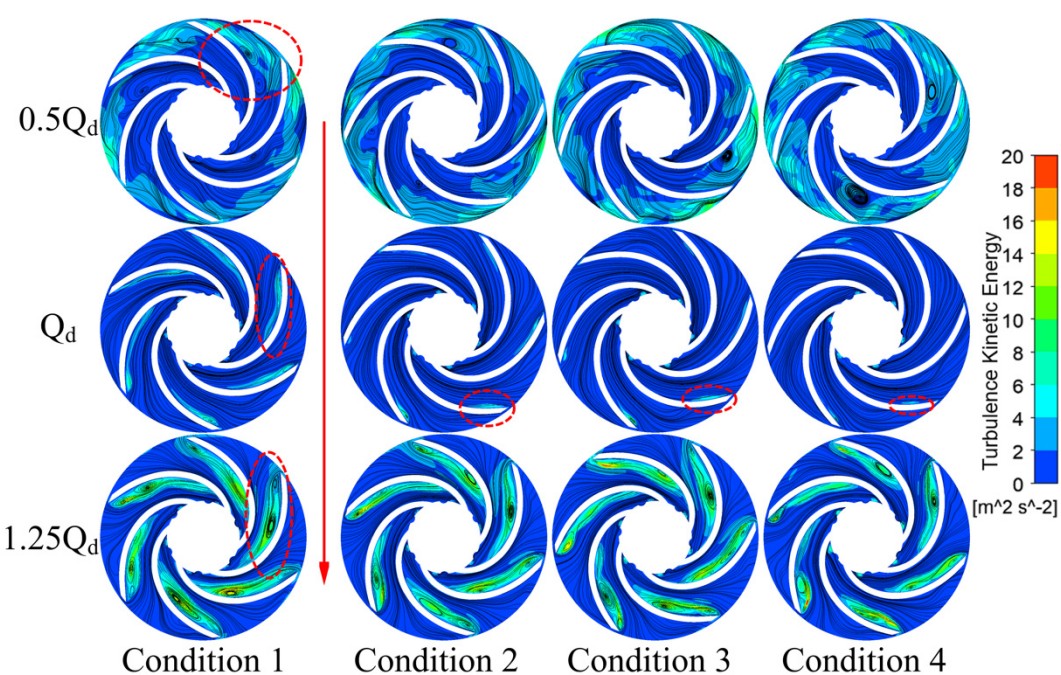

**Figure 13.** Turbulent kinetic energy distribution diagram for the impeller channel's radial section.

Figure 14 shows the turbulent kinetic energy distributions for the PAT's meridian section. At 0.5 $Q_d$, there was a high turbulent kinetic energy at the impeller's inlet, where the large vortex area blocked the channel. A larger increase in rate of flow led to a larger vortex cluster at the entrance and more chaotic streamlines at the exit, but the turbulent kinetic energy was evenly distributed without obvious energy loss. At $Q_d$, there were large vortex clusters for each set of working conditions at the junction of the flow channel exits, and there were also high turbulent kinetic energy regions at this junction. Comparing the four groups of working conditions, the working conditions with the greater increase in rate of flow had larger areas of high turbulent kinetic energy at the junction. This phenomenon corresponds to Figure 11c: for Conditions 1 to 4, greater flow rate increases led to lower maximum efficiencies. At 1.25 $Q_d$, the flow channel was filled with regions of high turbulent kinetic energy, and the streamline distributions were somewhat uneven. Comparing the four groups of unsteady working conditions, the working conditions with larger flow rate increases led to the flow channel having smaller high turbulence kinetic energy regions. This difference illustrates the phenomenon in Figure 11c. Although the highest efficiency point for Condition 4 was lower than that for other working conditions, as the flow rate increased, its efficiency declined the most slowly.

### 3.3. Stability Impact

A different increase in the rate of flow had different effects on the PAT's hydraulic performance. When the rate was less than 70 kg/s², the instantaneous efficiency change almost followed the efficiency curve for the steady conditions. Hydraulic performance is an important feature of PAT operation. It is also extremely important to study a PAT's stability for various working conditions. This study used the sliding mesh method to simulate the turbine's variable-condition operation, monitor the x-direction and y-direction fluid forces on the entire impeller, and obtain the runner's overall radial force vector through synthesis.

$$F_r = \sqrt{F_{rx}^2 + F_{ry}^2}. \tag{14}$$

In Equation (14), $F_{rx}$ and $F_{ry}$ are the components of the radial force in the x- and y-directions, respectively, $F_r$ is the total radial force.

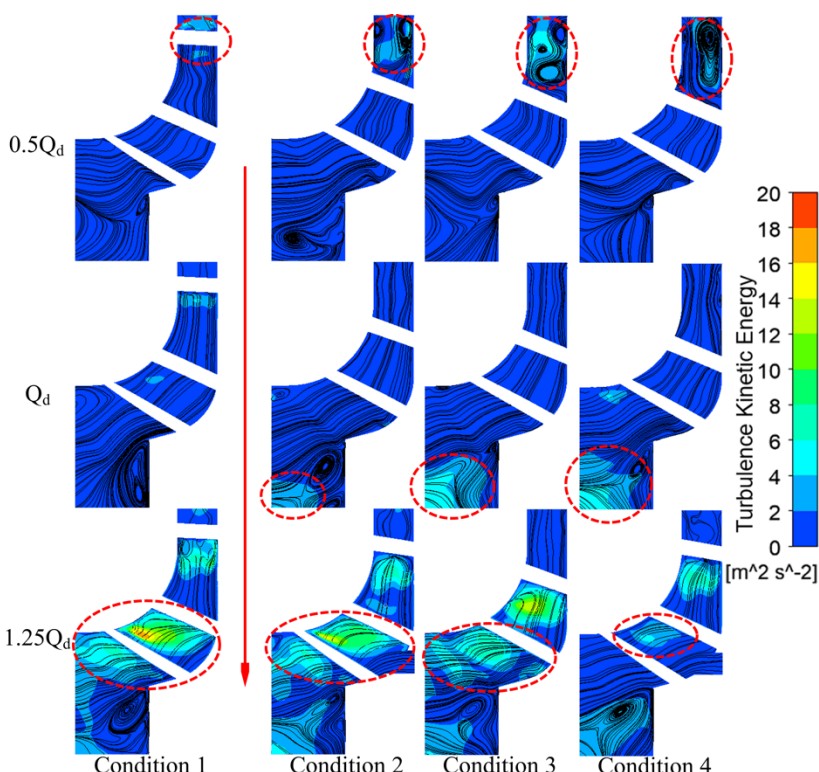

**Figure 14.** Turbulent kinetic energy distributions in the turbine's meridian section.

Figure 15a presents a polar diagram of the blade's radial force under constant flow conditions. For 0.5 $Q_d$, the turbulent flow field inside the impeller runner caused the blade's radial force to be large and change irregularly. For $Q_d$, the internal flow field was relatively stable, and the radial force was slightly larger than for 0.5 $Q_d$ and presented a certain periodicity. When the flow rate increased to 1.25 $Q_d$, the radial force continued to increase with increases in the flow rate, and it presented an obvious periodic change law, that is, there were six obvious fluctuations during a single rotation period.

Figure 15b,c shows polar diagrams for the blade's radial force under transient flow conditions. When the flow increased from 0.5 $Q_d$ to $Q_d$, the impeller's radial force decreased continuously and gradually presented a periodic change law. When $Q_d$ increased to 1.25 $Q_d$, the radial force for each set of working conditions gradually increased and had an obvious fluctuation law, and the working conditions with slower flow rate increases had larger blade radial force values. Compared with the constant flow condition, the radial force value of the transient flow conditions shows an obvious trend of first decreasing, and then increasing.

The solid lines in Figure 16 represent the instantaneous change curves of the impeller's axial force under transient flow conditions, and the dashed lines represent the axial force under different constant flow rates. The runner's axial force was positively correlated with the flow rate, and as the flow rate increased, the axial force increased more drastically. Different flow rate increases had a weaker impact on the axial force, indicating that the axial force was primarily related to the flow rate. At 0.5 $Q_d$, the axial forces for the transient flow conditions were close to those of the constant flow rate conditions. The axial force for a set of conditions with a larger flow rate increase was also slightly larger, but with increases in the flow rate, the axial force for different growth rate conditions gradually converged. For a constant $Q_d$, the axial force was significantly greater than for transient flow conditions, and the corresponding axial force values for each condition under 1.25 $Q_d$ were nearly equal. However, the fluctuations in the axial force for all conditions shared a common trend: the axial force fluctuated violently at low flow rates. At flow rates near $Q_d$, the fluctuations became gentle, then, as flow rate increased, the axial force amplitude

increased again. Figure 13 shows that the suction surface at the inlet of the impeller runner began to form a weak vortex near $Q_d$, which then developed and blocked the runner as the flow rate increased. The growth of this vortex is the reason the axial force gradually fluctuated away from stability.

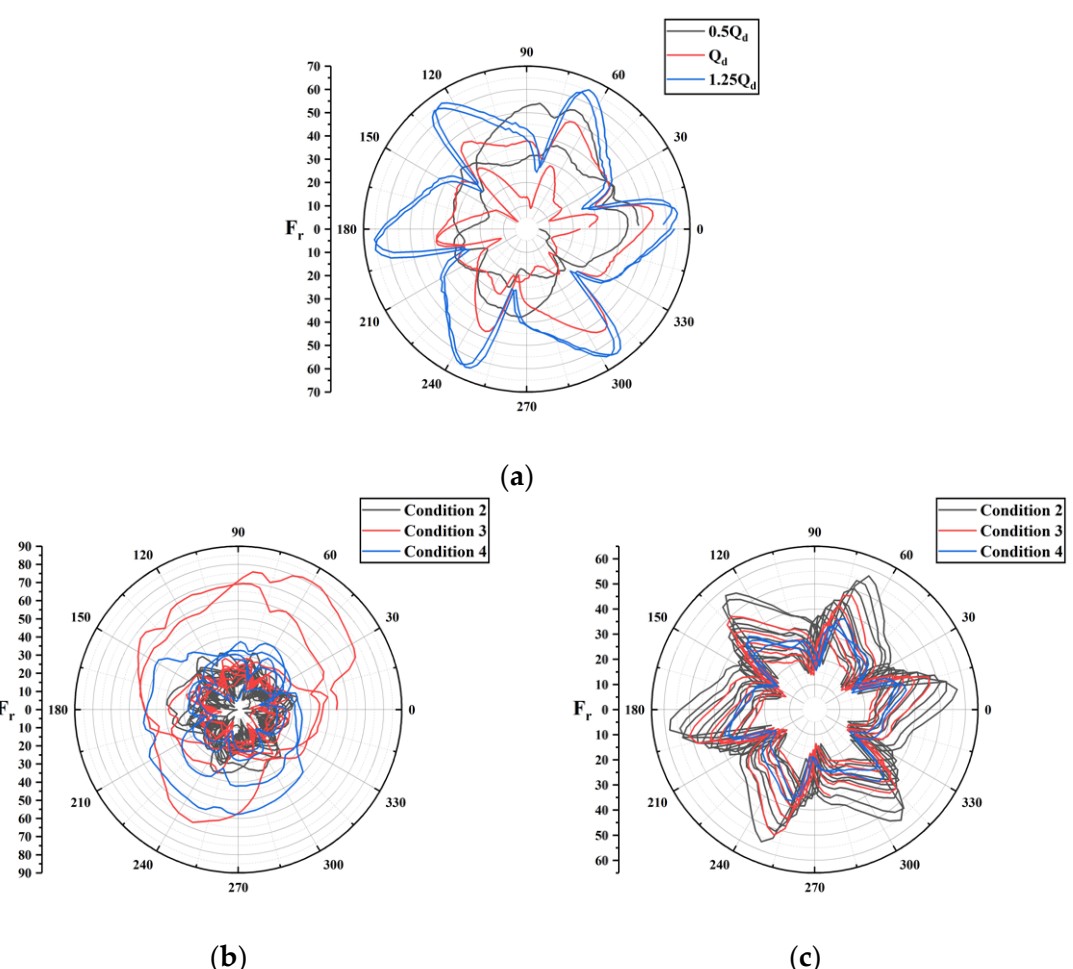

(a)

(b)                                                    (c)

**Figure 15.** Polar diagram of the blade's radial force under constant flow rate conditions (**a**), polar diagram of the blade's radial force under variable flow rate conditions from 0.5 $Q_d$ to $Q_d$ (**b**), and from $Q_d$ to 1.25 $Q_d$ (**c**).

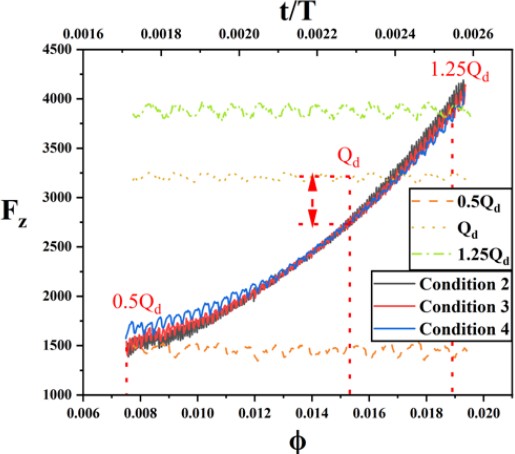

**Figure 16.** Instantaneous axial force of the runner for different working conditions.

Pressure pulsations in the turbine impact the safe and stable operation of the equipment. They are usually caused by random pressure pulsations caused by unstable secondary flow, backflow, wakes, vortexes, and cavitation; the pulsation of the rotor rotation, and the pulsation of the channel rotation. As shown in Figure 17, a series of monitoring points, P1–P8, were set in the turbine's volute to effectively monitor the pressure pulsation distribution in the flow member.

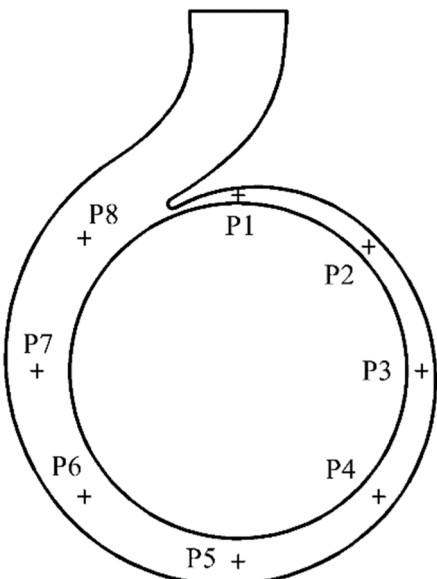

**Figure 17.** Distribution of the pressure pulsation monitoring points.

Figure 18 shows the pressure pulsations under stable working conditions. Affected by the rotor's periodic rotation, the pressures at the monitoring points in the volute experienced 12 peaks and troughs in two cycles, and the pressure pulsation changes during each cycle were similar. As the fluid flowed counterclockwise in the volute channel, from P8 to P1, the value of the average pressure and the amplitude of its pulsation continued to increase, indicating that a volute channel with a smaller cross-section can form greater pressure fluctuations. For 0.5 $Q_d$, the average pressure of each monitoring point was relatively close, but as the flow increased to $Q_d$ and 1.25 $Q_d$, there was a big difference between the average pressure of each monitoring point.

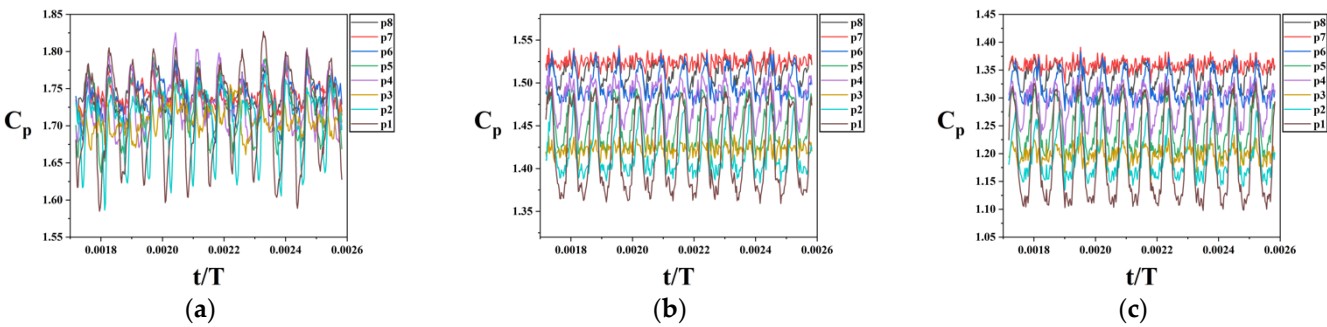

**Figure 18.** Pressure pulsations for Condition 1 at constant 0.5 $Q_d$ (**a**), $Q_d$ (**b**), and 1.25 $Q_d$ (**c**).

Figure 19 shows the pressure pulsations for the three sets of transient flow conditions. The relative pressure pulsation laws for each monitoring point were consistent with those of the stable conditions: the narrower the flow channel, the greater the pressure fluctuation. On the other hand, when the flow rate increased with time, the pressure amplitude at each monitoring point followed a consistent dynamic law: the pressure amplitude first decreased with increasing flow, then reached a minimum in the middle flow interval,

and finally increased with increasing flow. However, there were subtle differences for the three transient flow conditions. The flow interval corresponding to the minimum pressure amplitude for Condition 2 was approximately $0.81\ Q_d$, while the flow intervals corresponding to Conditions 3 and 4 were nearly $0.89\ Q_d$ and $Q_d$, respectively. That is, as the flow rate increased, the most stable pressure pulsation's flow range in the turbine volute also increased to the high flow region.

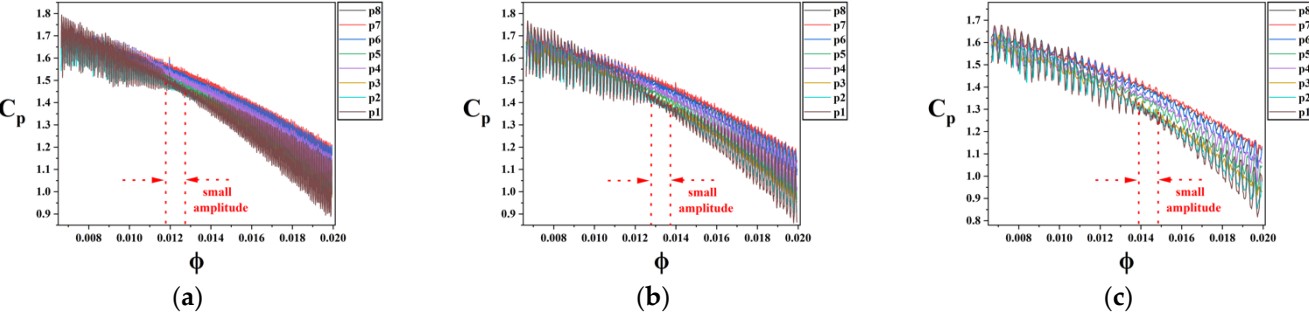

**Figure 19.** Pressure pulsations for Condition 2 (**a**), Condition 3 (**b**), and Condition 4(**c**).

## 4. Conclusions

PATs do not always have stable working conditions during actual operation. This research numerically studied the behavior of a PAT under transient flow conditions. Several notable conclusions were obtained from these simulation results.

The characteristic curves between transient flow conditions and steady working conditions are largely different. The actual efficiency is much lower than that for constant flow, due to the turbine stall phenomenon in the low flow rate range. A slow increase (large time derivative) of flow rate condition has a characteristic curve close to that of steady working conditions. Each transient flow condition's efficiency is closer to that for the constant conditions group after reaching the design flow rate.

The PAT's internal flow field diagram indicates that there is a small vortex cluster on the suction surface at the entrance of the impeller's flow channel under $Q_d$. The vortex continues to grow and block the flow channel as flow rate increases, which leads to a decrease in the turbine's energy conversion efficiency at high flows. Meanwhile, a larger increase in the rate of flow causes larger regions of high turbulent kinetic energy at the impeller's exit, making the PAT less efficient.

Compared with constant flow conditions, the radial and axial forces, and the pressure fluctuations of the impeller are much more violent under transient flow conditions with small and large flow rates, but are relatively stable near $Q_d$. This phenomenon occurs because of the development of the suction surface vortex in the impeller flow passage, which not only reduces the efficiency of the turbine, but also significantly affects its stability.

**Author Contributions:** Conceptualization, J.H. and X.S.; methodology, X.S.; software, X.H.; validation, X.H., C.C. and X.C.; formal analysis, X.H.; investigation, J.H.; resources, X.S.; data curation, Y.J.; writing—original draft preparation, X.H.; writing—review and editing, X.S.; visualization, K.W; supervision, K.W.; project administration, J.H.; funding acquisition, J.H. All authors have read and agreed to the published version of the manuscript.

**Funding:** This research was funded by National Natural Science Foundation of China, grant number 51906224.

**Institutional Review Board Statement:** Not applicable.

**Informed Consent Statement:** Not applicable.

**Data Availability Statement:** Not applicable.

**Conflicts of Interest:** The authors declare no conflict of interest.

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
