# Peer review of "Hydrodynamic Behavior of a Pump as Turbine under Transient Flow Conditions"

_processes, doi:10.3390/pr10020408_

Round 1

Reviewer 1 Report

Review of a paper entitled « Hydrodynamic behavior of a pump as a turbine under transient flow conditions” for publication in Processes by Jianxin Hu et al.

The paper proposes an interesting topic.

Nevertheless, it seems to the reviewer that some modifications of the content must be made by the authors before a potential publication.

Introduction:

  1. It seems necessary to explain what can be the various modes of operation (what is often called four-quadrants) of a given pump and the associated problems that can be encountered in each quadrant of operation.
  2. Some references must be added on that general subject, especially references to experimental and numerical approaches
  3. It must be the be clearly said by the authors what is the exclusive quadrant of operation of the present study.
  4. Regarding the paragraph on starting and stopping of a centrifugal pump, more references should be added and commented as that topic has been studied and published for a long time at least by US, Japanese and French researchers with especially a reference to spatial applications.
  5. Regarding the last paragraph, what means “unstable” for the authors? especially regarding transient operations
  6. Can the authors explain clearly what they call UDF?
  7. At the end of that section, it should be clearly said that the paper refer only to turbine operation

Numerical methods, models and validation:

  1. Section 2.1: what is the real interest of equations (1) to (5)? All notations are not defined; is there something new here?.
  2. Section 2.2:
    1. How are defined the extensions at inlet and outlet of the machine? What can be the influence of these domains on the quality (accuracy) of the numerical results?
    2. How is chosen the upper value of y+? what can be its influence on the quality of the results?
    3. Looking at figure 2, it is not so evident that hexahedral meshes are used.
    4. Can the authors say clearly that the effect of grid number has been studied only on one test case in steady operation in centripetal turbine. The term “fluctuations” of efficiency does not appear very adequate. Do the conclusion would be the same in other operating conditions and also in transient conditions? Some additional comments are necessary.
  3. Section 2.3:
    1. Has the roughness been compared to experimental values on the tested machine?
    2. Can the authors explain clearly why the MRF method is not available for transient simulations? And also explain what UDF means?
  4. Section 2.4:
    1. Figure 3a: indications are not easily readable
    2. Please add information about the machine: specific speed in pump operation and in turbine operation? Is the design a compromise between a pump design and a turbine design? Or is it a classical pump design?
    3. Table 2: the blade wrap angle is not clearly defined
    4. Page 6, line 162: it does not seem appropriate to the reviewer to speak of “errors”; it would be better to speak of differences between numerical and experimental values; the authors do not give any indication about experimental uncertainties as well as they do not give clear indication of what has been neglected for numerical simulation (side channels flows for example)
    5. Figure 6: difficult to read
  5. Section 3.1 and following:
    1. It appears that the proposed results have been obtained with transient calculations. The authors have to explain more extensively the numerical procedure (number of iterations during one time step, etc.).
    2. Can the authors explain how the four conditions (table 4) have been chosen. A discussion about quasi-steady conditions and fast or not rates of variations of flow conditions appear necessary. For example, some published papers on fast start up of pumps try to define some parameters to explain when some effects of the transient must be taken into account. That can be used for the present work, to add a scientific discussion to the figure 9 and other figures proposed in the next sections.

Reviewer 2 Report

This article investigates the transient operation of a PAT. The research topic is interesting and relevant.  The approach of combined experimental and CFD is good. The validation in Figure 6 looks good and provides some confidence in the CFD model.

Unfortunately there are several areas that need to be addressed before it its ready for publication.

  1. The grid independence shown in Table 1, should be presented in terms of a Richardson Extrapolation, and needs to have a wider range of points. The efficiency would be expected to vary more than 1 percentage point.
  2. Report the specific speed of the machine at best efficiency.
  3. Express results in a non-dimensional form. This makes the result more general and applicable to a wider range of applications. i.e. plot flow coefficient instead of flow rate,  head coefficient instead of head, power coefficient instead of power, dimensionless time.
  4. Case 1 and 2 should be named Condition 1 and 2 in Fig 7 a and b.
  5. The flow rate vs time in Fig 7 a) for Condition 2 is not linear, but it appears linear in Fig 8. This is a contradiction and is incorrect.
  6. Present the contours of pressure  in terms of pressure coefficient in Fig 10 (not pressure).
  7. The axis labels in Fig 16 and 17 are too small to read and should be plotted in terms of a pressure coefficient.
  8. Correct spelling and grammar errors (there are some that need fixing but it is mostly free of errors.

Round 2

Reviewer 1 Report

Thank you for the corrections

one remark : information about experimental uncertainties is not given

please read very carefully that new version, in order to correct errors regarding english language 

Author Response

We sincerely thank the reviewer for thoroughly examining our manuscript and providing very helpful comments to guide our revision. Please find the following responses to your comments and suggestions.

one remark : information about experimental uncertainties is not given

Reply: information about experimental uncertainties has been added (page 6 line 177-182).

please read very carefully that new version, in order to correct errors regarding English language 

Reply: we have carefully checked the manuscript and corrected relevant typographical errors and grammar errors.

Once again, we thank you for the time you put in reviewing our paper and look forward to meeting your expectations.

Reviewer 2 Report

The paper is substantially improved on the previous version. There are still typographical errors and minor grammar errors prior to publication,

Author Response

We sincerely thank the reviewer for thoroughly examining our manuscript and providing very helpful comments to guide our revision. Please find the following responses to your comments and suggestions.

The paper is substantially improved on the previous version. There are still typographical errors and minor grammar errors prior to publication.

Reply: we have carefully checked the manuscript and corrected relevant typographical errors and grammar errors.

Once again, we thank you for the time you put in reviewing our paper and look forward to meeting your expectations.